# Characteristics and Problems of Smart City Development in China

**Kaihui Huang [1,\*], Weijie Luo [1], Weiwei Zhang [2] and Jinhai Li [3]**

[1]  Advanced Institute of Information Technology, Peking University, Hangzhou 311215, China; wjluo@aiit.org.cn
[2]  School of Politics and International Relations, Lanzhou University, Lanzhou 730000, China; zhangww@lzu.edu.cn
[3]  Zhejiang Academy of Commerce, Hangzhou 311215, China; jwt1456@stu.ouc.edu.cn
\*  Correspondence: khhuang@aiit.org.cn

**Abstract:** The rapid expansion of urbanization both in scale and population leads to a series of serious urban diseases, which become a huge obstacle to the healthy and sustainable development of cities. To alleviate these problems and challenges, China launched a smart city construction program in the past decade and has taken the lead in smart city construction in the world. However, there is still a lack of reflection and summary on the practice of smart cities in China. Based on the definition and concept of smart city, this paper points out the internal and external driving factors of China's smart city development, then summarizes the four major characteristics of China's smart city construction practice, and explores the main problems existing in the process of China's smart city construction. Through the reflection and summary, we can facilitate development of smart cities in China, provide useful reference to urban planners and smart city practitioners in other countries and regions, and promote the healthy and sustainable development of cities.

**Keywords:** smart city; urbanization; sustainable development





## 1. Introduction

At present, more than 50% of the world's population lives in cities, and this figure will exceed 70% by 2050 [1]. Urbanization will continue to accelerate in the next 30 years. Cities, which occupy about 1% of the land area, consume 75% energy and 85% resources, and produce about 80% of greenhouse gas emissions of the world. With the continuous urbanization, cities will face more acute environmental pressure and contradiction between supply and demand of resources [1–3]. How to achieve sustainability while ensuring development is a major challenge for cities [4]. In the past 30 years, about 600 million people have moved from rural areas to cities [4], enabling China become the country with most cities containing population 1 million in the world. In the next 30 years, another 300 million people will move from rural areas to cities. However, rapid urbanization has also caused a series of urban diseases, such as traffic congestion, environmental pollution, contradiction between supply and demand of resources, etc. The problems and challenges in the process of urbanization have become one of the most urgent problems facing sustainable development in China and all countries around the world [5].

However, sustainable development increasingly depends on the effective management of cities [1]. With the goal of sustainable development, and livable and environmental protection, a smart city provides innovative solutions to the problems of urbanization [6,7]. It is of wide concern from global academia, industry, and government, and is considered an ideal solution to the challenges of urbanization [8]. It has become a strategic choice for most countries and regions to promote urbanization, improve the level of urban governance, solve the problems of big cities, and improve the quality of public services [9].

As a smart city has the potential to deal with the challenges and problems in the process of urbanization, after more than ten years of development in China, all large- and medium-sized cities in China have carried out smart city construction, making China

the world's largest smart city construction country, ranking first in both quantity and scale [10,11]. The practice of smart city construction in China provides valuable reference experience for the theory of smart city construction in other regions of the world.

The theory and practice of smart cities originated in developed countries, where urbanization began earlier and urban infrastructure was more perfect. However, in developing countries, which are the main force of current and future urbanization [12], the research on smart cities is obviously insufficient both in theory and practice [13]. At the same time, the current smart city research tends to be conducted from a single case that can intuitively observe the benefits and fragmentation, while there are few studies on the existing problems and negative impacts of smart cities [14,15]. As the world's largest developing country and smart city builder, China is a natural laboratory for observing smart cities and urbanization [16]. However, the research on China's smart cities mainly focuses on technology application [17], policy impact [15], construction effect [18], development path [19], etc., as well as the enlightenment of foreign smart city development experience for China [20,21]. China has its own unique characteristics, such as the urban–rural dual structure, concentrated urbanization, and rapid growth, which determine that China's smart city is different from other regions in the world [22]. A few studies have examined a single case or one of them, this breaks the integrity of the research to some extent [23–26]. In particular, the problems in the process of smart city construction in China are rarely reflected and summarized, and the causes of these problems are analyzed based on the actual situation and implementation effect in China [27].

In the past decade, the construction of smart cities in China has made certain achievements, showing typical Chinese characteristics, but at the same time, there are inherent problems [28]. Latecomers refer to excellent cases and lessons learned from smart cities to help improve city performances [29]. Based on the overall situation and construction practice of smart cities in China, this paper summarizes the characteristics of smart cities in China, provides more useful references for the construction of smart cities, and further improves the theory of smart cities from the practical level. This paper focuses on the analysis of the problems in the process of smart city construction and the reasons for the problems, so as to provide comparative guidance for the smart city construction in China and other regions. The research of this paper will promote the construction of smart cities in a more scientific and reasonable way for the government, urban planners, and ICT supply enterprises to construct smart cities with sustainable development.

## 2. Understanding the Connotation of Smart City

### 2.1. Definition of Smart City

Smart cities originated from the concept of smart growth in the context of the New urbanism movement in the United States in the 1980s and was produced in combination with information technology [30,31]. In the 1990s, "smart city" was first proposed in medias [32]. Due to different economic, geographical, and environmental conditions faced by different countries and regions, people have different perspectives and understanding of smart city and there is no unified definition [33,34]. Although many scholars expound some characteristics of smart city from the aspects of governance [35], technology [36], communication [37], transportation [38], population [39], environment [40], and so on, these concepts are subjective and one-sided and cannot be generally accepted [34]. On the one hand, the concept of a standardized smart city can avoid the confusion of smart city construction; on the other hand, it is possible to establish an objective and feasible smart city construction and evaluation standard. The widely accepted smart city framework is of great significance to solving the problems and challenges of global urbanization and realizing sustainable development [41–43]. Therefore, it is necessary to clarify the definition of smart city in order to better define the construction content of smart city and analyze the characteristics and problems of smart cities in China.

It is generally believed that a smart city is a path of urban development under the background of the development and application of information and communication technology

and the influence of smart growth urban planning, which is committed to improving the efficiency of urban management, achieving sustainable urban development, and improving the quality of life of urban residents [44]. The intelligent sustainable city group of ITU proposes that a smart city is an innovative city, which uses ICT and other means to improve the quality of life, urban operation and service efficiency, and competitiveness, while ensuring to meet the contemporary and future needs in economic, social, and environmental aspects [45]. This definition has a good guiding role in the development of smart city theory and the construction of different cities, which has been recognized by organizations and scholars such as the International Electrotechnical Commission. It is one of the widely accepted definitions in the current theoretical circles and official institutions [46].

### 2.2. Connotation of Smart City

Starting from the above definition, people have reached a consensus on the composition system of smart city, which is composed of people, economy, government, mobility, life, and environment [44]. On this basis, Ramaswami proposed five key construction fields of smart cities in economic opportunity, urban form, infrastructure, human wellbeing, and environmental protection [44,47]. Giffinger further pointed out six areas of smart cities: smart economy, smart citizens, smart governance, smart mobility, smart environment, and smart life (Figure 1) [48], which is known as the six dimensional theory of smart city and is recognized by most governments and scholars in the world in the smart city construction field [33,41,49], especially meeting the smart city demands in developing countries [29].

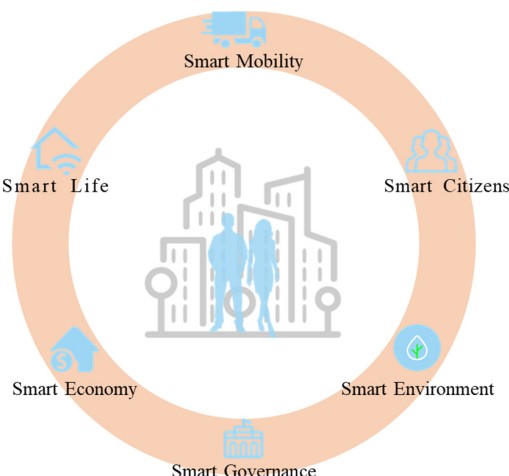

**Figure 1.** The six areas of smart city construction.

In short, understanding of definition and connotation of smart city varies due to differentiated development level and the objectives and situation of countries and their cities, especially in the development field of smart city, such as transportation, energy, and residents' health. However, smart city construction has many common characteristics: all using advanced technology to collect, process, and analyze data; different subsystems within the whole system realizing the real-time information exchange without obstacles; life of urban residents being more intelligent; and the urban ecological environment being more livable and sustainable. This is also the meaning of the above concepts [50,51].

### 3. Driving Factors of Smart City Development in China

### 3.1. External Driving Force of Smart City Development

Government policy support is an important external driving force for the development of smart cities. On one hand, the construction of a smart city is a realistic demand for China to deal with urban problems and promote the modernization of urban governance system and governance capacity, which requires the government to formulate systematic support policies on guidance and promotion [52]. On the other hand, smart city construc-

tion involves new infrastructure fields such as 5G, which is an effective investment for local governments to promote GDP growth through public infrastructure construction. Therefore, local governments should formulate support policies from the perspective of input–output, increasing quality and scale of resources input in smart city construction and helping form large-scale construction effect under specific institutional order.

In 2012, the Ministry of Industry and Information Technology announced the first batch of smart city pilot areas. In 2013, the State Council issued "Opinions on Promoting Consumer Information Technology Expenditure and Expanding Domestic Demand" ("Opinions") to encourage pilot cities to introduce policies to support market development and use information resources to participate in the infrastructure construction of smart cities. In the same year, "National New Urbanization Plan (2014–2020)" clearly needed to promote the construction of smart cities and unify material resources, information resources, and intellectual resources of urban development. Subsequently, seven Ministries and Commissions jointly issued "Guiding Opinions on Promoting Healthy Development of Smart Cities" (hereinafter referred to as Guidance), pointing out that the development of smart cities should be people-oriented, based on their own needs, market-oriented, and avoid unnecessary government intervention. The "2015 Government Work Report" pointed out that smart technologies and smart cities are two major priorities for China's development and breakthroughs should be made in fields of "digital and smart technology". So far, policy supporting the system of China's smart city construction has taken shape, marking the arrival of China's smart city construction boom. According to Google trend's search statistics of smart city keywords in China, search peaks all appeared within one year after releasing the above policy documents. We also found that the number of related research papers in China has increased since 2012, which largely reflects the policies' influence on promoting the construction of smart cities (Figures 2 and 3).

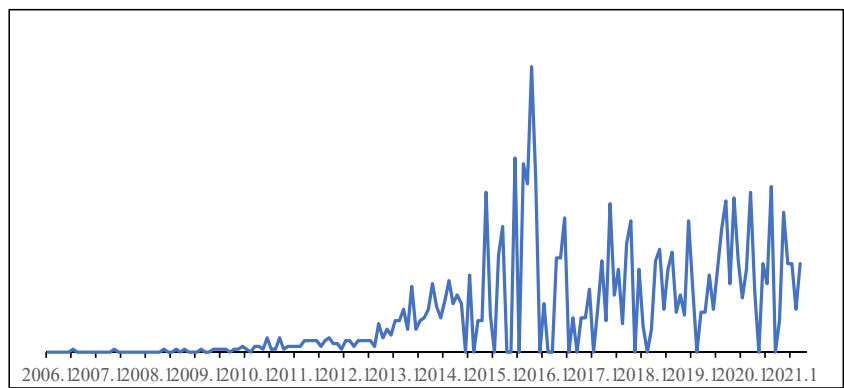

**Figure 2.** China's smart city search trend from 2006 to 2021 (data source: Google trend).

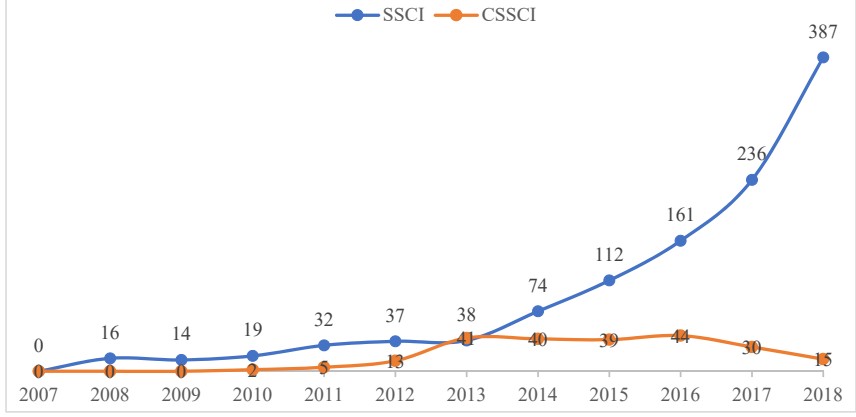

**Figure 3.** Statistics of SSCI and CSSCI smart city papers (revised from [53]).

The publicity and promotion of ICT enterprises is also an important driving force for the construction of smart cities. As direct beneficiary of smart cities, ICT enterprises play an important role in infrastructure construction and technology application and the promotion of smart cities. IBM, which puts forward the concept of "smart city", has held 100 smart city forums around the world, including 22 in China in 2009, and successfully promoted the concept. The idea of smart city is widely accepted in China. Nanjing, Shenyang, Chengdu, and Kunshan have carried out strategic cooperation with IBM and smart city construction projects have been launched one after another [54].

### 3.2. Internal Driving Force of Smart City Development

Urbanization is a core feature of smart cities. Economic opportunities are key driving forces for the development of cities; infrastructure is the main driving factor; creating fairness for all citizens is a fundamental purpose; and a good ecological environment is an important guarantee. However, the problems faced by urbanization in economy, environment, people's livelihood, governance, mobility. and life inevitably require intensive intelligent solutions, which is the internal driving force for promoting smart cities [55]. For the main problems, development level and development stages are different among regions, the development focus of smart cities is different. In the six areas, the weight of intelligent environment is far higher than other factors. The survey of opinions on smart city construction of urban residents shows that residents in both developed countries and developing countries think that environment is most important, and most residents think that smart city construction should provide comfortable living conditions and working atmosphere for residents [51]. Environment is closely related to residents of each city, providing a greener and healthier environment could significantly reduce health problems caused by environment (Figure 4).

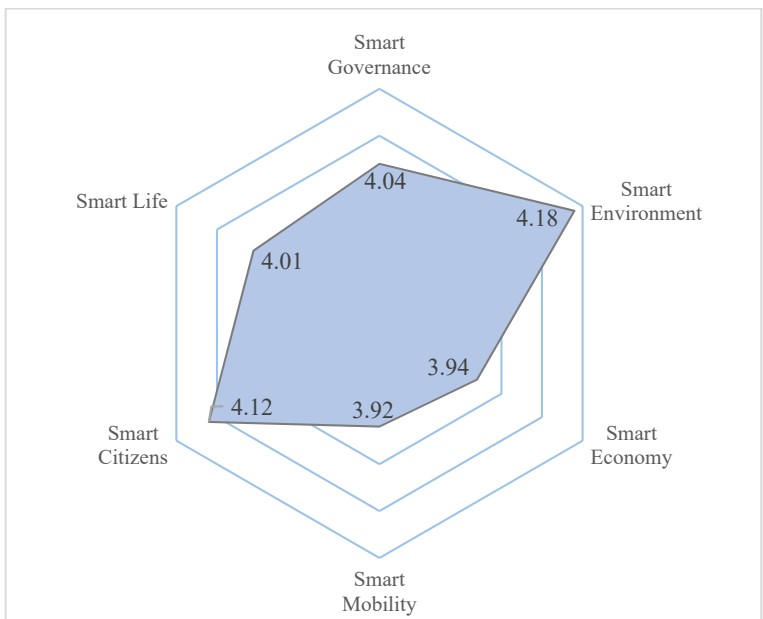

**Figure 4.** The weight of six driving factors on smart city construction in developing countries (revised from [51]).

Empirical analysis of smart city construction in developing countries shows that the contribution of different fields to smart city construction is also quite different. In developing countries, smart environment is the main driving force of smart city; smart residents and smart governance are also key factors to smart city construction [56].

## 4. Characteristics of Smart City Development in China

### 4.1. The Top-Down Construction Mode

The theory and practice of smart city originates from independent behavior of local cities in Western countries, construction sponsored by local cities, where roles of central government are relatively vague. Different from Western countries, central government plays a leading role in the construction of smart cities in China, showing the characteristics of top-down construction.

As the connotation and construction fields of smart cities coincide with goal of promoting modernization of national governance system and governance capacity on construction path, central government takes smart cities as a national strategy. The construction area, construction field, and evaluation standard are issued by superior government through administrative instructions and implemented by local government. In the construction process, government takes the lead, construction and innovation institutions participate together, government integrates the system and resources of the whole city, focuses on efficiency, quality, and economic advantages of urban management, emphasizes the use of advanced technology on providing better support for operation of the city, and comprehensively solves the problems of urban development. In construction mode, it integrates technology and policy. Government plays an important decision-making role. Due to huge investment, the government dominates in financing, development, and operation.

In the short term, the top-down model is convenient for large-scale promotion and implementation, achieving goals more quickly. However, due to lack of public consultation, participation, and accountability supervision mechanisms, this model has received more and more reflection. The practice of smart city construction believes that people's participation is very necessary and of great value. By fully taking people's opinions into consideration, their most urgent needs can be effectively solved [57]. The bottom-up model driven by people's demands can hear more voices of more people during policy-making and change the model of citizens' effective participation in urban governance [58]. On one hand, people interact with service providers through various devices and technologies through data as they are the beneficial group of smart city service; on the other hand, citizens' participation can better solve challenges in the implementation process as data creators.

In the long run, the combination of top-down mode and bottom-up mode is a better choice [59]. Top-down construction has characteristics of centralized decision-making and decentralized implementation, which to a certain extent is not conducive to cities' independent potential. The local construction and fields are homogenized according to the instructions of the higher authorities. In practice, China's smart city construction mainly focuses on smart government, smart transportation, and smart public services, but innovation-driven smart services have not received enough attention. Citizens are significant factors to enhance a city's innovation ability. Bringing citizens into a city's innovation system can not only design programs that meet the needs of the people, but also produce products, services, or new or innovative ideas that are generally suited for needs of the society [60].

### 4.2. Technocentrism

Technology is the main driving force of urban development [61]. Smart cities are based on the development of ICT represented by the Internet of Things, big data, 5G, artificial intelligence, and cloud computing. By exploring the efficacy of data collection, we can analysis and achieve more efficient city management and services, which could promote industrial transformation. In the construction of smart cities in China, a series of policy documents also emphasize the basic role of technology, e.g., the "Opinions" focus on the development of hardware and technology infrastructure. Smart cities are also regarded as the application of ICT in urban planning, construction, management, and operation. Promoting the application of ICT is the main idea of smart city construction in China, which reflects the characteristics of technocentrism.

ICT is regarded as the path to explore future city scenarios [62]. Under the concept of technocentrism, urban activities are deconstructed into system models constructed by different business layers and data streams [63]. As an invisible leading force, technology follows the logic of continuous upgrading and optimizing system performance to promote a city's evolution. The role of information infrastructure is to optimize the operation process of the system, improve the efficiency of the system, and realize accurate control, intervention, and quantitative prediction of cities.

Technology is the key to building a smart city, but it is not enough to make a city smart. On one hand, the technology-oriented smart city ignores social acceptance of technology in decision-making processes; on the other hand, it takes enterprises as goals to maximize interests and lacks a humanistic concept, forming a technology-oriented "enterprise smart city" [64]. Due to a lack of social and cultural inclusiveness, technology-centric and enterprise-driven smart cities have many disadvantages. Technocentrism focuses on the application and promotion of technology and does not pay enough attention to people-oriented application services and innovation-driven sustainable development, which is directly manifested as "emphasizing infrastructure construction and neglecting service application", resulting in a lack of market foundation for commercial operation [65]. The integration of technology and society is particularly important for public acceptance. Otherwise, smart city construction will face major risks, such as Songdo in South Korea and Masdar in the UAE. At the same time, lack of overall coordination and technocentrism aggravates data islands and digital divide [53].

Technology plays a fundamental role in management, resource allocation, and efficiency improvement of smart city. It is the basis of building smart ecosystems to meet the needs of urban economy, society, and environment and realize the sustainable development of city and people. Through technology application enabling system and infrastructure integration, ICT brings opportunities for innovation and change of urban intelligence, but technology is only a means, not the purpose of making city better. Real smart cities need to start from the city and its social problems, technology-centered needs must turn to people-centered, balancing relationships between technology, innovation, people, society, innovation, culture, and environment [65,66].

### 4.3. Huge Market Scale

At present, more than 800 cities in China are promoting the planning and construction of smart cities, which is the largest scale in the world and forms multiple smart city clusters. As mentioned above, such a large-scale smart city construction cannot be separated from strong promotion of policies. In 2012, the Ministry of Housing and Urban-Rural Development issued the first batch of 90 pilot lists of smart cities. Since then, relevant departments have issued a series of documents and determined a number of pilot areas for smart city construction. These policies have boosted smart cities to national strategies.

Driven by national strategies and policies, China's smart cities have sprung up rapidly, forming several smart city clusters in Yangtze River Delta and Pearl River Delta. A huge market scale and volume provide broad market and development opportunities for technology companies, such as Huawei, Alibaba, and Tencent, which have signed strategic cooperation framework agreements with more than 300 cities. According to Deloitte's statistics, the current investment scale of smart city construction in China is more than US$ 25.9 billion, of which IT investment from 2011 to 2018 was more than US$ 300 billion, market scale accounted for more than 22% of the world, and the proportion will be further increased. It is estimated that the investment scale will exceed US$ 40 billion in 2023 (Figure 5).

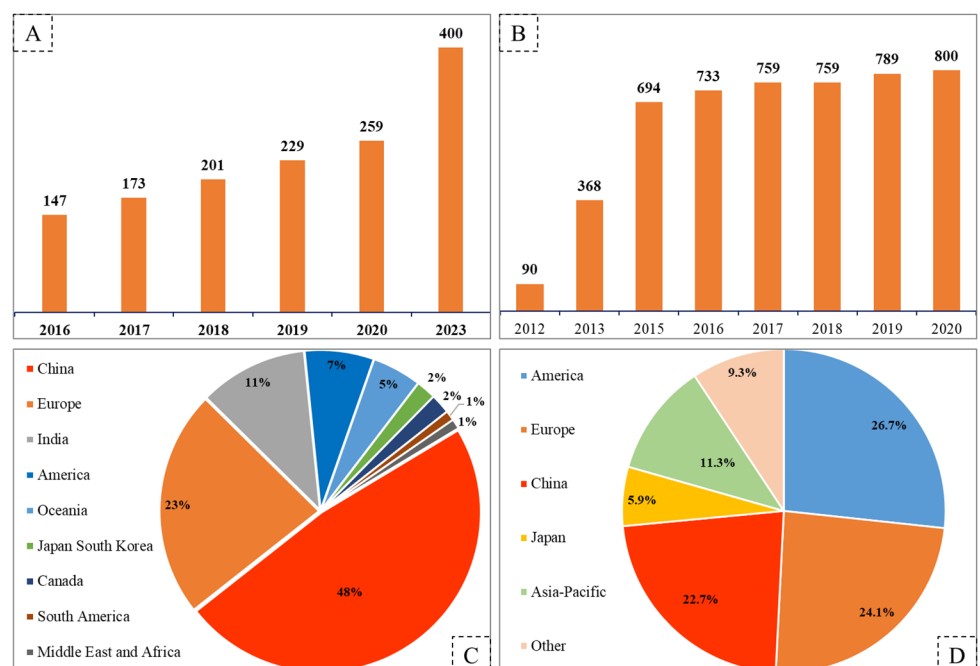

**Figure 5.** Smart city construction in China. (**A**), the number of smart cities constructed in China in recent years (data source: Qianzhan Industry Research Institute); (**B**), investment scale of smart cities in China; (**C**), proportion of the number of smart cities constructed in major countries; (**D**), proportion of smart city investment scale in major countries around the world.

Such large-scaled smart city construction is also closely related to China's rapid urbanization and economic growth. There is a certain correlation between the growth rate of China's urbanization rate and GDP growth (Figure 6). Economic development creates more employment opportunities for populations transferred to cities, so that transferred rural migrants could gain a foothold in cities and towns. Since the Reform and Opening up, China's urbanization process has been deepened, with characteristics of concentration in big cities and metropolitan areas. The early areas of smart city exploration are mainly concentrated in eastern coastal cities.

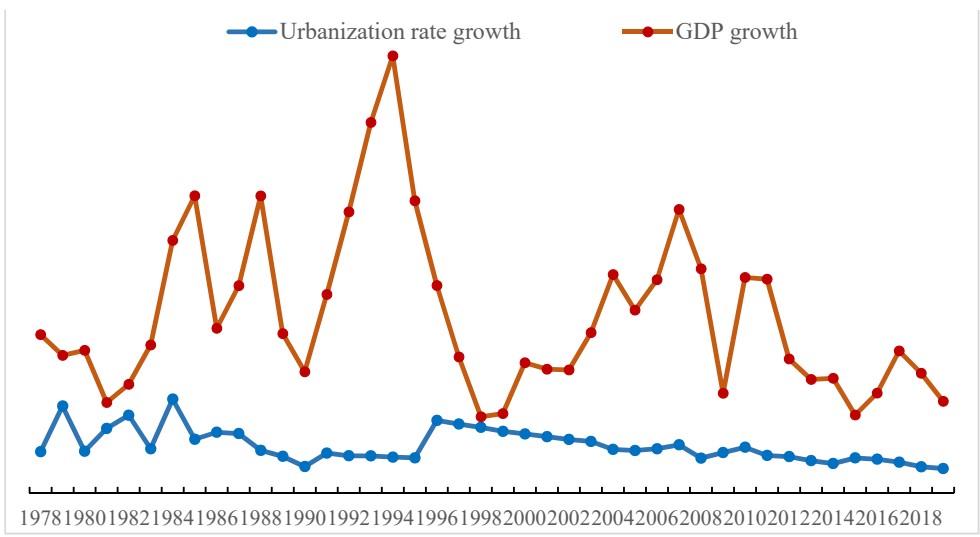

**Figure 6.** China's population, urbanization rate, and GDP growth rate changes since the Reform and Opening up (data source: China Statistical Yearbook).

### 4.4. Construction Foundation and Effect Differentiation

In essence, smart cities are based on informatization and deepened and upgraded with the help of ICT. However, due to different development levels and informatization degrees in different regions, there is a huge gap between small cities and developed cities such as Beijing, Shanghai, Guangzhou, and Shenzhen in terms of technical ability and informatization degree. There is also a big disparity in effects.

Due to differences in level of information development between regions, there are also great differences in construction foundation. In 2013, nearly half of China's first batch smart city construction list was concentrated in eastern coastal developed areas, while the number in the western region, with relatively weak economic foundation, only accounted for one-third of that of the eastern region. In the following years, central and western regions continued to strengthen investments in the IT field under the promotion of the "Internet +" policy; the city's informatization level has been improved. With the improvement of urban optical fiber, 4G network, WiFi, cloud computing data center, Internet of Things, and other facilities, the second and third round of smart city construction pilot areas are relatively evenly distributed in eastern, central, and western cities (Figure 7). Although smart cities in central and western regions have been developed and improved to a great extent, compared with developed regions, there is a certain gap between smart city construction achievements and quality in central and western regions, e.g., smart city development in Beijing, Shanghai, Guangzhou, and Shenzhen has entered a new stage, but most areas are still in the initial or transitional stage.

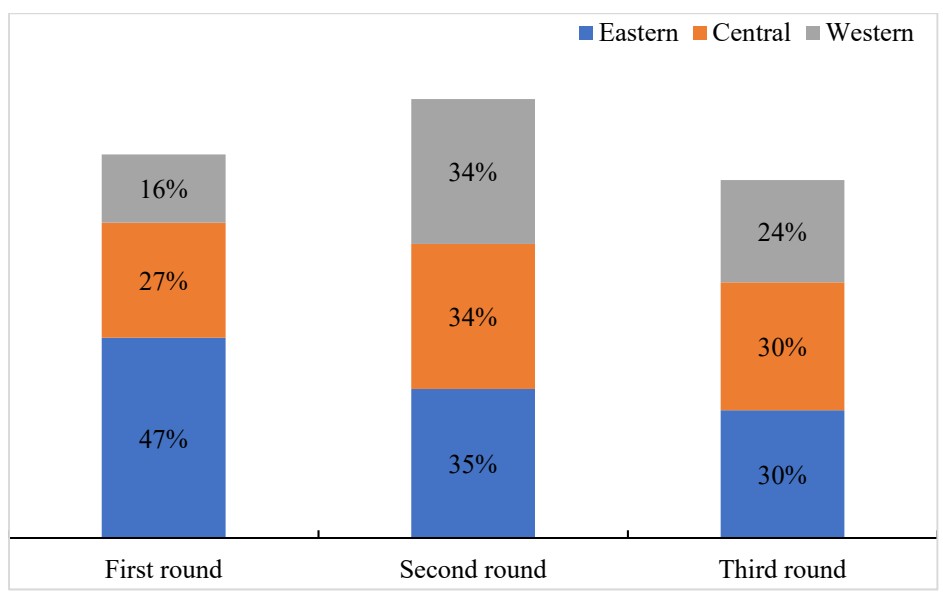

**Figure 7.** The proportion of different regions in three rounds of smart city construction in China [28].

Due to different foundations, there are great differences in smart city construction effects, as well as construction priorities. Eastern coastal cities emphasize social governance, such as strengthening the public service capacity of environmental governance, health care, etc.; for western cities, more attention is paid to information facilities to improve urban innovation and economic vitality. Based on analysis of 32 major smart cities' construction in China, it was found that they can be divided into four categories: 1. High level of intelligence and good effect, such as coastal cities with good development foundation, strong economic strength, technological innovation, and management ability; there is large investment in social capital and special funds. Local and international companies have good innovation cooperation relationship with the government in talent, technology, standard, and management, which have a high degree of intelligence and efficiency; 2. Low intelligence and good effect. Due to a relatively weak economy, technology companies are

attracted to participate in technology cooperation through preferential technology policies and strengthening the construction of information infrastructure, but there is still a lack of R&D capacity, mainly in central cities; 3. Low level of intelligence and weak effect. Due to geographical location and economic reasons, technical cooperation in Northwest China is limited, while the economic development is weak, emerging industries scale is small, key technology and R&D abilities are poor, and the capital source of smart city is heavily dependent on finance; 4. High level of intelligence and insignificant effect. This kind of city has strong economic power and a large number of high-tech companies, but the construction effect is not outstanding [16]. Each city implements policies according to different goals. As time goes by, the gap in effectiveness of smart city construction will further widen.

## 5. Existing Problems

Predecessors have given different opinions on problems existing in the development of smart cities in China at this stage. Most scholars believe that China has problems such as weak top-level design, data islands, closed industrial ecology, waste of resources, and scare public participation [67,68]. Based on field visits and observations, this article believes that there are mainly the following six problems according to connotation and construction fields of smart cities.

### 5.1. Lack of Plans and Goals

As a complex giant system, smart cities involve various groups of governments, enterprises, and residents, which highly integrate industries, cities, and residents and needs investment in large amounts of capital, technology, and human resources. It requires clear plans and top-level designs before smart city construction starts.

At present, smart city construction schemes are mostly provided by technology-oriented construction enterprises and often lack sufficient attention understanding of actual local needs and major challenges. In addition, construction goals are not clear, decision-makers have a biased understanding of smart cities and lack information governance capabilities. Governments passively accept bidding proposals from enterprises, which leads to many plans imitating each other and repeated functions, making smart city construction not only lack pertinence, but also unable to form complementarity between cities.

The unclear top-level plans and objectives make smart city construction focus on one or several aspects and subsystems, e.g., many regions deploy a large amount of information infrastructure in transportation, energy, education, medical care, governance, and other aspects, but this cannot form an organic smart city as a whole, causing imbalanced systems including infrastructure, governance, people, management, economy, and environment [69]. The lack of top-level plans leads to unclear content of smart city construction, resulting in waste of resources, low efficiency of construction, and hidden dangers of information security.

Due to the lack of top-level plans and clear objectives, infrastructure mainly focuses on technology application and promotion. Smart city infrastructure in most areas of China is seriously insufficient, especially the infrastructure related to residents' health and urban equity [15]. Smart cities are for people's better survival and development; China also emphasizes people-oriented construction evaluation standard, people-oriented should firstly protect the "right to life", public infrastructure is to provide citizens with clean air, clean water, and safe food. Public infrastructure is not only related to smart city construction effects, but also related to the ecology and residents' health [70]. To solve these problems, it is necessary to explore new paths of urban infrastructure construction with more intelligent and intensive technologies. The ISO-TC268 sustainable development City Technical Committee takes smart city infrastructure construction as assessment content to measure and evaluate construction effects.

### 5.2. More Environmental Protection Should Be Combined

Sustainable urban growth must ensure development is inclusive, safe, resilient, and sustainable. These characteristics make a city competitive; otherwise the city's development will face failure. Environmental problems not only endanger residents' health, but also hinder the efficiency of economic operation. Economic loss caused by environmental problems in China is as high as 10% of GDP every year [71]. Therefore, smart cities must consider the sustainability of urban development and pay attention to environmental protection in the construction process.

Environmental problems caused by rapid urbanization process are severe challenges to sustainable development. Cities produce less than 10% of their total food needs, consume 70% of world's total energy, and produce more than 75% of the garbage, which puts heavy pressure on the environment. China is top in the number of cities with a population over one million in the world and is the largest carbon emitter in the world. As a signatory to the Paris Agreement, China has committed to increasing its share of non-fossil fuel energy responsibility to more than 20% by 2030 and achieving carbon neutrality by 2060. As the country with the fastest urbanization rate growth in the world, China will need at least 1.6–1.9 billion square meters of commercial and residential space every year in the next 10 years to meet the transportation and housing needs of the urban population. The continuous rapid urbanization will further aggravate energy consumption, water and air pollution, and toxic waste emissions, and increase burden on the environment. Therefore, it is necessary to seek a sustainable urban environment [72].

Although most smart cities in China put forward the construction goal of smart ecology in the top-level design, in practice, the attention and investment of smart cities to the environment is not enough [73]. The current goal of smart environment is mainly the government's administrative demand, which focuses on environmental monitoring and emergency management, and lacks active practice to realize the systematic protection of cities and environments from a larger level. According to the effect evaluation of 44 major smart cities in China from five dimensions: people, environment, economy, governance, and information facilities, the score of smart environment is also at the lowest level (Figure 8). Cities are a part of the natural ecosystem, focusing on the urban system alone, but lacks the research and protection of the interaction between the city and the whole ecosystem. Its role and effect are limited, which is not conducive to the sustainable development of the city.

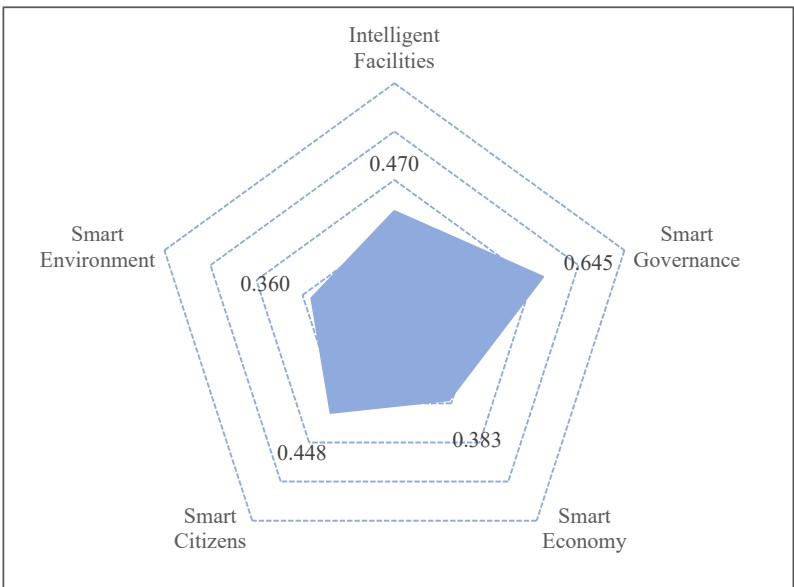

**Figure 8.** The smart indicators of major cities in China (revised from [15]).

### 5.3. Homogenization of Construction Content

In the past 20 years of research and construction practice at home and abroad, smart cities have seldom taken into account local economic and cultural differences as a common goal of urban construction. Economic and cultural differences in different regions will have an impact on technology and systems [74]. Cities' geographical characteristics, culture, and knowledge ecosystem have vast degrees of shaping innovation ability and technology acceptance, which will affect the effectiveness of smart cities. Therefore, smart city construction should choose schemes suitable for local characteristics.

China has a vast territory and there is a big gap in regional economic development levels. The differences between the east and the west and the north and the south are obvious and there is a growing trend. The factors' endowment structure, such as population, per capita income of residents, and industrial base, varies from region to region, which has a significant impact on the effect of smart city construction. According to the theory of comparative advantage, prices of capital, labor, technology, and other resource elements vary greatly in different regions [75]. Different cities have different economic structures and social spatial development models. Economically developed areas adopt a capital-intensive and technology-intensive model, with labor costs being higher. In economically underdeveloped areas, it is quite the opposite. This difference not only affects demand of industries for urban infrastructure, but also determines the direction of urban development and economic structure. If the construction of a smart city cannot combine local actual factor endowment structure but only copies experiences of other cities, it will distort normal value of resource elements, cause the wasting of resources and low operation efficiency, and have a negative impact on economic development.

Differences in regional economic, political, and cultural factors also lead to differences in technological needs among cities. Empirical analysis shows that by singularly judging from the perspective of urban population scale and economic development level, the difference of the effect efficiency of smart city construction on economy, urban governance, and ecology is significant. For cities with a large population, social governance and ecological protection effects are better; for cities with a smaller population, it is the opposite; big cities have better information technology foundation, which is more conducive to changing urban governance mode and easier to play economic agglomeration effect, while it is difficult for small cities to achieve an agglomeration effect and scale effect due to weak information technology [71].

### 5.4. Insufficient Economics of Construction and Operation

Due to large capital investment and the long-term nature of smart city construction, economic construction and operation mode is a big problem faced by smart cities. From practice at home and abroad, construction and operation mode for which the government is directly responsible cannot meet the capital and service needs of smart cities and operation efficiency is low. Although many places have explored the construction and operation of PPP mode, enthusiasm of social capital participation is generally not high due to uncertain short-term commercialization prospects.

The economic problems of construction and operation are mainly due to the following reasons: firstly, the public attribute of smart city infrastructure. At present, smart city construction in China is mainly invested in by government, which is a big financial burden for most local governments. Secondly, the underlying information and communication infrastructure belongs to public goods in the construction system of smart city, which is basic and irreplaceable and has no significant profit prospects. Finally, data application services are insufficient. As mentioned above, the realization of data opening and sharing is the basis of realizing commercial operation. Only through data mining, analysis, and serving needs of different types of users of government, enterprises, and the public can value-added innovation and development be realized.

The operation of smart cities is a long-term problem, but there is still a lack of a clear business model. The construction fields of a smart city should be classified. In

some areas of public fundamental services, social capital cannot benefit and generally has no enthusiasm to enter. Government can invest in construction by means of public expenditure. For the parts that can yield commercial operation and investment return prospect, they could be opened to the public, such as intelligent parking and intelligent building, thus providing models and explorations for other value-added services. On an operational level, multi-channel financing and operation mode are explored and the government pays more attention to the formulation and implementation supervision of standards, laws, and top-level design under the guidance of marketization.

### 5.5. Data Fusion and Open Barriers

In the process of smart city construction, how to achieve data fusion is a difficult problem in current smart city construction. Data islands are a common problem for smart cities. Most people believe that administrative classification hinders information sharing. The fundamental reason for information islands is that the lack of top-level design and technology dominates one or several fields, which leads to unbalanced system performance and conflict of heterogeneous data systems, causing resource monopoly [17,73].

In fact, the data integration problem is mainly due to technical obstacles. On the one hand, smart city application services should be based on large amount of data generated by citizens, which are usually mastered by different government departments and agencies, stored in different formats and managed by different protocols. There is widespread monopoly of resources, which leads to interoperability obstacles. It is very difficult for different departments to share sensitive data, especially for some governments that lack sufficient information technology capabilities, whose data islands are more serious. On the other hand, all kinds of information infrastructures produce different kinds and structures of data. There are no uniform standards for realizing data formats, especially IoT devices that produce a large amount of unstructured data. Current mining and analysis tools are still in the initial stage of development; China is seriously short of core technologies such as information systems and data management and can only rely on foreign technologies [39]. Therefore, there are still many technical difficulties in how to realize data fusion and dynamic interactive operation at different levels such as GIS, BIM, and IoT.

Lack of uniform industry standards, construction standards, and evaluation standards leads to a serious waste of data resources and low construction efficiency. The data generated by smart cities are increasing day by day. Data with spatial and temporal characteristics have become an important asset of smart cities, which play an important role in the construction and development of smart cities. There is an urgent need for effective data mining and use [76]. As the carrier for information sharing, exchange, and coordination services, data should be open to establish interaction among government, enterprises, and the public, so as to provide intellectual support for smart city innovation and solve the service and application needs of smart cities. As innovation has typical characteristics of high risk and high failure rate from the perspective of an innovative four-helix model and the implementation effect of the European Living Lab, data openness not only allows more stakeholders to participate widely in innovation, improves innovation efficiency, and improves people's satisfaction with smart city construction, but also has important significance for smart city application services.

### 5.6. Development Level Needs to Be Improved

Smart cities are not only a current urban phenomenon, but also a global urban movement, which is regarded as a testing ground for innovation and an indicator to evaluate urban innovation. The world's major cities compete globally, improving their development level through smart cities and creating a global city brand to attract resources and talents [77].

Although the number and scale of China's smart cities account for a high proportion in the world, there is a big gap between China and developed countries in terms of international smart city score. EasyPark selects 500 cities from the United Nations Human

Development Index and Prosperity Index, focusing on the use of technology to create smart cities, and evaluates them from seven aspects: transportation and mobility, sustainability, governance, innovative economy, digitization, and living standard. Among the top 100 of the Global Smart City Index 2019, only four cities in China are on the list: Taipei, Hong Kong, Shanghai, and Beijing. The scores of Beijing and Shanghai in mainland China are relatively low. IESE has established a theoretical model of Cities in Motion Index (CIMI) from perspectives of governance, urban planning, technology, environment, international projection, social cohesion, human capital, mobility, transportation, and economy. The 2020 report shows that only Shanghai and Beijing are listed in the top 100 cities in mainland China (Table 1). Although smart city assessment has a variety of evaluation indicators, European smart city ranking is considered to be a more effective tool [77].

**Table 1.** Ranking of major smart cities in China.

| City | EasyPark the Smart Cities Index 2019 | IESE CIMI |
|---|---|---|
| Taipei | 25 [1] | 27 [2]/70.78 [3] |
| Hong Kong | 87 | 10/76.04 |
| Shanghai | 93 | 58/62.38 |
| Beijing | 99 | 84/56.27 |

[1,2] rank. [3] score.

Previous empirical analysis also finds that China's cities are still at a relatively low level, including intelligent infrastructure, governance, people, economy, and environment. At the same time, due to the large difference of intelligence between cities in China, the economic gap between cities has a trend of further increasing.

## 6. Conclusions

This paper takes China's smart city construction as the research object. Based on the concept of smart city and taking the concept of smart city with high acceptance as the starting point, we defined the concept of smart city. Smart governance, smart economy, smart environment, smart citizens, smart life, and smart mobility are the main components of smart city construction. China's smart city is inseparable from the promotion of national policies and enterprises, which is an important external driving factor. The problems and challenges faced by urbanization are the endogenous driving force and fundamental reason for the development of smart city.

This paper argues that smart cities in China mainly present the following remarkable characteristics: 1. driven by the goal of modernization of governance system and governance capacity, it presents a top-down construction mode; 2. focus on ICT application and implementation, which becomes a technology-centered model; 3. rapid urbanization provides the basis for smart city construction, which occupies high market share globally in both scale and volume; 4. there is a big gap in construction effect due to infrastructure and regional economic differences.

After more than ten years of development, smart cities in China have achieved a series of results. Different cities have carried out beneficial explorations based on their own actual situations. However, many cities blindly carry out smart city construction due to the requirements of superior policies, and do not combine with the actual local needs, have unclear planning and objectives, and copy the construction experience of other regions, resulting in serious homogenization of construction contents, failure to form dislocation development, and complement each other among different regions. These problems further distort the value of resources and jointly lead to the economic imbalance of smart city construction and operation. At the same time, local urban planners have weak awareness of environmental protection and insufficient understanding of the challenges of urban sustainable development. Blindly copying the experience and practices of other regions directly leads to the homogenization of construction content. In addition, technical problems hinder the integration and opening of data, which brings obstacles to market-

oriented operation and economic benefits. In general, China's smart city construction is still at the stage of development and upgrading; there is still a big gap between some smart cities and the actual needs of urban development.

At present, China is in a leading position in quantity and scale of smart cities, but construction quality of smart cities needs to be improved, which requires us to attach importance to the development connotation of smart cities. The construction of smart cities ultimately serves the needs for people, so we should establish a people-oriented concept. Considering the sustainable development demand of cities, the relationship between cities and environment should be paid attention and environmental protection should be integrated into the important construction field. At the same time, the development should combine resource endowment and industry characteristics, in accordance with the actual situation of the region against the background of urbanization, by keeping economy abreast with security, to highlight local features of smart city construction.

**Author Contributions:** The conceptualization, methodology, formal analysis, investigation, data curation, writing, review, and editing were done by K.H.; W.L.; W.Z. and J.L. All authors have read and agreed to the published version of the manuscript.

**Funding:** This research received no external funding.

**Institutional Review Board Statement:** Not applicable.

**Informed Consent Statement:** This study did not involve humans.

**Data Availability Statement:** The data presented in this study are available on request from the corresponding author.

**Conflicts of Interest:** The authors declare no conflict of interest.

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
