# Peer review of "Characteristics and Problems of Smart City Development in China"

_smartcities, doi:10.3390/smartcities4040074_

Round 1

Reviewer 1 Report

Your work is timely and interesting. However I have some suggestions and comments for the improvement in your work as the followings:

  1. Abstract should not be written in points.
  2. Your paper has formatting issues, Journal fonts are not used. Please correct it.
  3. Please mention the source of Fig.1 because this figure is previously used in many publications.
  4. In heading 4.3 please start it with a capital letter.
  5. Please recheck your conclusion, Smart City is not theoretically a robust construct rather it is more into application.

Author Response

Dear reviewers:

Thank you very much for your valuable suggestions on this manuscript. Your suggestions are very pertinent and important and point out the deficiencies of the article. Your suggestions have been discussed by all authors, combined with the actual situation of the article, and responded to one by one. The following will explain and explain the changes one by one in accordance with the suggestions of the article.

“Comments and Suggestions for Authors:

Your work is timely and interesting. However I have some suggestions and comments for the improvement in your work as the followings”

Point 1: Abstract should not be written in points.

Response 1:

According to your opinion, with reference to the peer articles in this journal, the summary has been revised, and the point by point format has been deleted.

Point 2: Your paper has formatting issues, Journal fonts are not used. Please correct it.

Response 2:

The manuscript has been rechecked and the format has been modified. The standard template provided by the editorial department (smartcities-template.dot (live.com)) is used here.

Point 3: Please mention the source of Fig.1 because this figure is previously used in many publications.

Response 3:

Based on the definition of smart city, analyzing the connotation of smart city and referring to the results of different researchers, this paper believes that the construction scope of smart city should focus on six aspects: smart economy, smart governance, smart life, smart mobile, smart environment and smart residents. In order to more intuitively display the construction scope of smart city, this paper draws Figure 1 independently. For similar drawings, Building intelligent system for smart cities: issues, challenges and approvals and The rise of the smart city.

Point 4: In heading 4.3 please start it with a capital letter.

Response 4:

The problems here have been corrected and other problems in the article have also been modified.

Point 5: Please recheck your conclusion, Smart City is not theoretically a robust construct rather it is more into application.

Response 5:

The conclusion of this paper has been checked and modified again. Specifically modify the content and summarize the main content of the manuscript, that is, first introduce the connotation of smart city and the driving factors of China's smart city, then analyze the characteristics and main problems of China's smart city, and finally introduce the shortcomings of the article and the research focus in the next step.

Reviewer 2 Report

This manuscript focuses on a topic of sure interest for the readership of the Smart Cities journal. However, it has some unclear issues and has some weaknesses. Some of these weaknesses are a lack of justification for key assumptions in the analysis, others are a lack of attention to detail incorrectly describing the analysis that is being performed. I describe these weaknesses in more detail below. Therefore, despite the very interesting issue covered, the manuscript needs to be carefully revised. My main concerns are the following:

- The abstract should be more concrete, precise, sharp and critical.

- The introduction should be clearer. The literature review is a major drawback of the paper. It is not carried out well. The authors throw in different concepts/theories, thus, making the storyline difficult to follow/read. You should engage more with the relevant academic literature. It is analysed superficially, without a detailed analysis of relevant literature. There is no clear problem statement. In addition, you should indicate what are the innovative contributions of your manuscript to science. Therefore, I suggest the authors rewrite the introduction and literature review, making it explicit and clearer to read/understand.

- what are the limitations of your study?

- Please emphasise the contribution and implication of the paper. It is necessary that there is at least a clear contribution to the field of knowledge (e.g., an innovative approach to a methodology, etc.).

- Major grammar and punctuation errors can be found throughout the text and need to be corrected.

Author Response

Dear reviewer:

Thank you very much for your valuable suggestions on this manuscript. Your suggestions are very pertinent and important and point out the deficiencies of the article. Your suggestions have been discussed by all authors, combined with the actual situation of the article, and responded to one by one. The following will explain and explain the changes one by one in accordance with the suggestions of the article.

Point 1: The abstract should be more concrete, precise, sharp and critical.

Response 1: The summary has been revised and adjusted again. This paper mainly introduces the shortcomings of China's smart city research under the background of urbanization, then introduces the main contents of the article, and finally states the research significance and value of this paper.

Point 2: The introduction should be clearer. The literature review is a major drawback of the paper. It is not carried out well. The authors throw in different concepts/theories, thus, making the storyline difficult to follow/read. You should engage more with the relevant academic literature. It is analysed superficially, without a detailed analysis of relevant literature. There is no clear problem statement. In addition, you should indicate what are the innovative contributions of your manuscript to science. Therefore, I suggest the authors rewrite the introduction and literature review, making it explicit and clearer to read/understand.

Response 2:

  1. Introduction this part has been rewritten. The ideas of this part are as follows: firstly, it introduces the challenges and problems faced by urban sustainable development under the background of global urbanization. China is carrying out the largest planned urbanization process in the world, and the challenges and problems of sustainable development are as urgent as those in other parts of the world. In this context, smart cities are fully launched in China and have achieved a series of results. Next, the paper points out the main problems existing in the research of smart city, that is, although smart city is more urgent and important in developing countries, the current research is missing and insufficient, especially for China. China is the largest smart city construction country in the world, with its own characteristics. Finally, it introduces the value and significance of this paper. The research on China's smart city will provide a reference for the construction of smart city in other regions, avoid problems and deficiencies, promote the better development of smart city and contribute to the sustainable development of the city.
  2. In the literature review part of introduction, this paper is carried out step by step according to the above writing ideas, and analyzes and quotes relevant research work.

Point 3: what are the limitations of your study?

Response 3:

  1. The smart city objects studied in this paper mainly focus on China's large and medium-sized cities, and there are deficiencies in the research on County cities. In China, smart city is to serve the urbanization process. From the perspective of policy evolution, China's urbanization process initially emphasizes the urbanization of big cities, but China has the typical characteristics of urban-rural dual structure. This policy guidance intensifies the imbalance and separation of urban and rural development, so urbanization is adjusted to emphasize the coordinated development of urban and rural areas. In this paper, the investigation and data of County cities are not fully mastered, which will be the focus of my research in the next stage.
  2. This paper analyzes the main problems existing in the smart city. This study can not provide detailed information of various indicators in each dimension to enrich the decision-makers which information is more important. It also analyzes the background and causes of the problems, but does not give specific suggestions on the solutions to some problems.
  3. This study only uses quantitative analysis. Although these quantitative analysis is based on the analysis of the results of smart city construction, further research is needed to provide scientific empirical analysis results.

Point4: Please emphasise the contribution and implication of the paper. It is necessary that there is at least a clear contribution to the field of knowledge (e.g., an innovative approach to a methodology, etc.).

Response 4

Firstly, it systematically summarizes the main characteristics of smart cities in China in recent ten years, and analyzes these common problems. China is the largest smart city construction country in the world. It is very meaningful and necessary to systematically introduce the construction of smart cities in China.

Secondly, although the theory and practice of smart city began in developed countries, the systematic summary and analysis of so many construction practice cases in China will better promote the development of smart city theory, whether positive or negative.

Finally, as the largest developing country, China's experience and existing problems in smart city construction will provide valuable reference experience for other parts of the world, especially for developing countries and regions facing large-scale urbanization. China's case is very important and can provide valuable reference experience for latecomers to avoid possible problems.

Point 5: Major grammar and punctuation errors can be found throughout the text and need to be corrected.

Response 5

Native speakers of English are invited to discuss and revise the language and problems in the article with the author.

Round 2

Reviewer 2 Report

Dear Authors,

Thank you for taking into consideration my remarks. In my opinion, the manuscript can be published.